# Learning by Minimizing the Sum of Ranked Range

**Shu Hu**
University at Buffalo, SUNY
shuhu@buffalo.edu

**Yiming Ying**
University at Albany, SUNY
yying@albany.edu

**Xin Wang**
CuraCloud Corporation
xinw@curacloudcorp.com

**Siwei Lyu**
University at Buffalo, SUNY
siweilyu@buffalo.edu

## Abstract

In forming learning objectives, one oftentimes needs to aggregate a set of individual values to a single output. Such cases occur in the aggregate loss, which combines individual losses of a learning model over each training sample, and in the individual loss for multi-label learning, which combines prediction scores over all class labels. In this work, we introduce the sum of ranked range (SoRR) as a general approach to form learning objectives. A ranked range is a consecutive sequence of sorted values of a set of real numbers. The minimization of SoRR is solved with the difference of convex algorithm (DCA). We explore two applications in machine learning of the minimization of the SoRR framework, namely the AoRR aggregate loss for binary classification and the TKML individual loss for multi-label/multi-class classification. Our empirical results highlight the effectiveness of the proposed optimization framework and demonstrate the applicability of proposed losses using synthetic and real datasets.

## 1 Introduction

Learning objective is a fundamental component in any machine learning system. In forming learning objectives, we often need to aggregate a set of individual values to a single numerical value. Such cases occur in the aggregate loss, which combines individual losses of a learning model over each training sample, and in the individual loss for multi-label learning, which combines prediction scores over all class labels. For a set of real numbers representing individual values, the ranking order reflects the most basic relation among them. Therefore, designing learning objectives can be achieved by choosing operations defined based on the ranking order of the individual values.

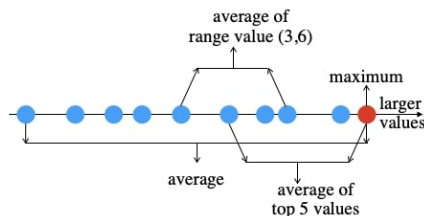

Figure 1: *Illustrative examples of different approaches to aggregate individual values to form learning objectives in machine learning. The red dot corresponds to a potential outlier.*

Straightforward choices for such operations are the average and the maximum. Both are widely used in forming aggregate losses [35, 30] and multi-label losses [22], yet each has its own drawbacks. The average is *insensitive* to minority sub-groups while the maximum is *sensitive* to outliers, which usually appear as the top individual values. The average top-$k$ loss is introduced as a compromise between the average and the maximum for aggregate loss [11] and multi-label individual loss [12]. However, it dilutes but not exclude the influences of the outliers. The situation is graphically illustrated in Fig.1.

In this work, we introduce the sum of ranked range (SoRR) as a new form of learning objectives that aggregate a set of individual values to a single value. A ranked range is a consecutive sequence of sorted values of a set of real numbers. The SoRR can be expressed as the difference between two sums of the top ranked values, which are convex functions themselves. As such, the SoRR is the difference of two convex functions and its optimization is an instance of the difference-of-convex (DC) programming problems [17]. The non-smoothness of the SoRR function and the non-convex nature of the DC programming problem can be efficiently solved with the DC algorithm (DCA).

We explore two applications in machine learning of the minimization of the SoRR framework. The first is to use the *average* of ranked range (AoRR) as an aggregate loss for binary classification. Unlike previous aggregate losses, the AoRR aggregate loss can completely eliminate the influence of outliers if their proportion in training data is known. Second, we use a special case of SoRR as a new type of individual loss for multi-label, the TKML loss, which explicitly encourages the true labels in the top $k$ range. The new learning objectives are tested and compared experimentally on several synthetic and real datasets[1]. The main contributions of this work can be summarized as follows:

- We introduce SoRR as a general learning objective and show that it can be formulated as the difference of two convex functions, which leads to an efficient solution based on the DC algorithm.
- Based on SoRR, we introduce the AoRR aggregate loss for binary classification, and establish its classification calibration with regards to the optimal Bayes classifier.
- We also introduce the TKML individual loss for multi-label learning, which is another special case of SoRR, and show that the TKML loss is a lower bound of the conventional multi-label loss.
- We empirically demonstrate the robustness and effectiveness of the proposed AoRR, TKML, and their optimization framework on both synthetic and real datasets.

## 2 Sum of Ranked Range

For a set of real numbers $S = \{s_1, \cdots, s_n\}$, we use $s_{[k]}$ to denote the *top-$k$ value*, which is the $k$-th largest value after sorting the elements in $S$ (ties can be broken in any consistent way). Correspondingly, we define $\phi_k(S) = \sum_{i=1}^{k} s_{[i]}$ as the *sum of the top-$k$* values of $S$. For two integers $k$ and $m$, $1 \leq m < k \leq n$, the $(m, k)$-ranked range is the set of sorted values $\{s_{[m+1]}, \cdots, s_{[k]}\}$. The sum of $(m, k)$-ranked range ($(m, k)$-SoRR) is defined as $\psi_{m,k}(S) = \sum_{i=m+1}^{k} s_{[i]}$, and the average of $(m, k)$-ranked range ($(m, k)$-AoRR) is $\frac{1}{k-m} \psi_{m,k}(S)$. It is easy to see that the sum of ranked range (SoRR) is the difference between two sum of top values as, $\psi_{m,k}(S) = \phi_k(S) - \phi_m(S)$. Also, the top-$k$ value corresponds to the $(k-1, k)$-SoRR, as $\psi_{k-1,k}(S) = s_{[k]}$. Similarly, the median can also be obtained from AoRR, as $\frac{1}{\lceil \frac{n+1}{2} \rceil - \lfloor \frac{n+1}{2} \rfloor + 1} \psi_{\lfloor \frac{n+1}{2} \rfloor - 1, \lceil \frac{n+1}{2} \rceil}(S)$.

In machine learning problems, we are interested in the set $S(\theta) = \{s_1(\theta), \cdots, s_n(\theta)\}$ formed from a family of functions where each $s_i(\theta)$ is a convex function of parameter $\theta$. We can use SoRR to form learning objectives. In particular, we can eliminate the ranking operation and use the equivalent form of SoRR in the following result. Denote $[a]_+ = \max\{0, a\}$ as the hinge function.

**Theorem 1** *Suppose $s_i(\theta)$ is convex with respect to $\theta$ for any $i \in [1, n]$, then*

$$\min_{\theta} \psi_{m,k}(S(\theta)) = \min_{\theta} \left[ \min_{\lambda \in \mathbb{R}} \left\{ k\lambda + \sum_{i=1}^{n} [s_i(\theta) - \lambda]_+ \right\} - \min_{\hat{\lambda} \in \mathbb{R}} \left\{ m\hat{\lambda} + \sum_{i=1}^{n} [s_i(\theta) - \hat{\lambda}]_+ \right\} \right]. \quad (1)$$

*Furthermore, $\hat{\lambda} > \lambda$, when the optimal solution is achieved.*

The proof of Theorem 1 is in the Appendix A.1. Note that $\psi_{m,k}(S(\theta))$ is not a convex function of $\theta$. But its equivalence to the difference between $\phi_k(S(\theta))$ and $\phi_m(S(\theta))$ suggests that $\psi_{m,k}(S(\theta))$ is a difference-of-convex (DC) function, because $\phi_k(S(\theta))$ and $\phi_m(S(\theta))$ are convex functions of $\theta$ in this setting. As such, a natural choice for its optimization is the DC algorithm (DCA) [27].

To be specific, for a general DC problem formed from two convex functions $g(\theta), h(\theta)$ as $s(\theta) = g(\theta) - h(\theta)$, DCA iteratively search for a critical point of $s(\theta)$ [33]. At each iteration of DCA, we first form an affine majorization of function $h$ using its sub-gradient at $\theta^{(t)}$, *i.e.*, $\hat{\theta}^{(t)} \in \partial h(\theta^{(t)})$,

and then update $\theta^{(t+1)} \in \text{argmin}_\theta \left\{ g(\theta) - \theta^\top \hat{\theta}^{(t)} \right\}$. DCA is a descent method without line search, which means the objective function is monotonically decreased at each iteration [32]. It does not require the differentiability of $g(\theta)$ and $h(\theta)$ to assure its convergence. Moreover, it is known that DCA converges from an arbitrary initial point and often converges to a global solution [17]. While a DC problem can be solved based on standard (sub-)gradient descent methods, DCA seems to be more amenable to our task because of its appealing properties and the natural DC structure of our objective function. In addition, as shown in [28] with extensive experiments, DCA empirically outperforms the gradient descent method on various problems.

To use DCA to optimize SoRR, we need to solve the convex sub-optimization problem

$$\min_\theta \left[ \min_\lambda \left\{ k\lambda + \sum_{i=1}^n [s_i(\theta) - \lambda]_+ \right\} - \theta^T \hat{\theta} \right].$$

This problem can be solved using a stochastic sub-gradient method [5, 29, 31]. We first randomly sample $s_{i_l}(\theta^{(l)})$ from the collection of $\{s_i(\theta^{(l)})\}_{i=1}^n$ and then perform the following steps:

$$\theta^{(l+1)} \leftarrow \theta^{(l)} - \eta_l \left( \partial s_{i_l}(\theta^{(l)}) \cdot \mathbb{I}_{[s_{i_l}(\theta^{(l)}) > \lambda^{(l)}]} - \hat{\theta}^{(t)} \right),$$

$$\lambda^{(l+1)} \leftarrow \lambda^{(l)} - \eta_l \left( k - \mathbb{I}_{[s_{i_l}(\theta^{(l)}) > \lambda^{(l)}]} \right)$$

(2)

**Algorithm 1:** DCA for Minimizing SoRR

**Initialization:** $\theta^{(0)}$, $\lambda^{(0)}$, $\eta_l$, and two hyperparameters $k$ and $m$

**for** $t = 0, 1, ...$ **do**
    Compute $\hat{\theta}^{(t)}$ with Eq.(3)
    **for** $l = 0, 1, ...$ **do**
        Compute $\theta^{(l+1)}$ and $\lambda^{(l+1)}$ with
        Eq.(2)
    **end**
    Update $\theta^{(t+1)} \leftarrow \theta^{(l+1)}$
**end**

where $\eta_l$ is the step size. In Eq.(2), we use the fact that the sub-gradient of $\phi_m(S(\theta))$ is computed, as

$$\hat{\theta} \in \partial \phi_m(S(\theta)) = \sum_{i=1}^n \partial s_i(\theta) \cdot \mathbb{I}_{[s_i(\theta) > s_{[m]}(\theta)]}, \tag{3}$$

where $\partial s_i(\theta)$ is the gradient or a sub-gradient of convex function $s_i(\theta)$ (Proof can be found in the Appendix A.2)[2]. The pseudo-code of minimizing SoRR is described in Algorithm 1.

## 3 AoRR **Aggregate Loss**

SoRR provides a general framework to aggregate individual values to form learning objective. Here we study in detail of its use as an aggregate loss in supervised learning problems and optimizing it with the DC algorithm. Specifically, we aim to find a parametric function $f_\theta$ with parameter $\theta$ that can predict a target $y$ from the input data or features $x$ using a set of labeled training samples $\{(x_i, y_i)\}_{i=1}^n$. We assume that the individual loss for a sample $(x, y)$ as $s_i(\theta) = s(f(x;\theta), y) \geq 0$. The learning objective for supervised learning problem is constructed from the aggregate loss $\mathcal{L}(S(\theta))$ that accumulates all individual losses over training samples, $S(\theta) = \{s_i(\theta)\}_{i=1}^n$. Specifically, we define the AoRR aggregate loss as

$$\mathcal{L}_{aorr}(S(\theta)) = \frac{1}{k-m} \psi_{m,k}(S(\theta)) = \frac{1}{k-m} \sum_{i=m+1}^k s_{[i]}(\theta).$$

If we choose the $\ell_2$ individual loss or the hinge individual loss, we get the learning objectives in [25] and [14], respectively. For $m \geq 1$, we can optimize AoRR using the DCA as described in Section 2.

The AoRR aggregate loss is related with previous aggregate losses that are widely used to form learning objectives.

- the *average loss* [36]: $\mathcal{L}_{avg}(S(\theta)) = \frac{1}{n} \sum_{i=1}^n s_i(\theta)$;
- the *maximum loss* [30]: $\mathcal{L}_{max}(S(\theta)) = \max_{1 \leq i \leq n} s_i(\theta)$;
- the *median loss* [21]: $\mathcal{L}_{med}(S(\theta)) = \frac{1}{2} \left( s_{\left[ \lfloor \frac{n+1}{2} \rfloor \right]}(\theta) + s_{\left[ \lceil \frac{n+1}{2} \rceil \right]}(\theta) \right)$;
- the *average top-$k$ loss* (AT$_k$) [11]: $\mathcal{L}_{avt-k}(S(\theta)) = \frac{1}{k} \sum_{i=1}^k s_{[i]}(\theta)$, for $1 \leq k \leq n$.

The `AoRR` aggregate loss generalizes the average loss ($k = n$ and $m = 0$), the maximum loss ($k = 1$ and $m = 0$), the median loss ($k = \lceil \frac{n+1}{2} \rceil$, $m = \lfloor \frac{n+1}{2} \rfloor - 1$), and the average top-$k$ loss ($m = 0$). Interestingly, the average of the bottom-$(n-m)$ loss, $\mathcal{L}_{abt-m}(S(\theta)) = \frac{1}{n-m}\sum_{i=m+1}^{n} s_{[i]}(\theta)$, which is not widely studied in the literature as a learning objective, is an instance of the `AoRR` aggregate loss ($k = n$). In additional, the robust version of the maximum loss [30], which is a maximum loss on a subset of samples of size at least $n - (k - 1)$, where the number of outliers is at most $k - 1$, is equivalent to the top-$k$ loss, a special case of the `AoRR` aggregate loss ($m = k - 1$).

Using the `AoRR` aggregate loss can bring flexibility in designing learning objectives and alleviate drawbacks of previous aggregate losses. In particular, the average loss, the maximum loss, and the $\text{AT}_k$ loss are all influenced by outliers in training data, which correspond to extremely large individual losses. They only differ in the degree of influence, with the maximum loss being the most sensitive to outliers. In comparison, `AoRR` loss can completely eliminate the influence of the top individual losses by excluding the top $m$ individual losses from the learning objective.

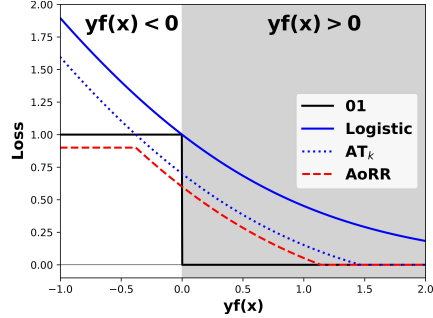

Figure 2: *The* `AoRR` *loss and other losses interpreted at the individual sample level. The shaded area over 0.00 loss corresponds to data/target with the correct classification.*

In addition, Traditional approaches to handling outliers focus on the design of robust *individual losses* over training samples, notable examples include the Huber loss [13] and the capped hinge loss [23]. Changing individual losses may not be desirable, as they are usually relevant to the learning problem and application. On the other hand, the `AoRR` loss introduces robustness to outliers when individual losses are aggregated. The resulting learning algorithm is more flexible.

The robustness to outliers of the `AoRR` loss can be more clearly understood at the individual sample level, with fixed $\lambda$ and $\hat{\lambda}$. We use binary classification to illustrate with $s_i(\theta) = s(y_i f_\theta(x_i))$ where $f_\theta$ is the parametric predictor and $y_i \in \{\pm 1\}$. In this case, $y_i f_\theta(x_i) > 0$ and $y_i f_\theta(x_i) < 0$ corresponds to the correct and false predictions, respectively. Specifically, noting that $s_i(\theta) \geq 0$, we can rearrange terms in Eq.(1) to obtain

$$\mathcal{L}_{aorr}(S(\theta)) = \frac{1}{k-m} \min_{\lambda>0} \max_{\hat{\lambda}>\lambda} \sum_{i=1}^{n} \left\{ [s_i(\theta) - \lambda]_+ - [s_i(\theta) - \hat{\lambda}]_+ \right\} + k\lambda - m\hat{\lambda}. \qquad (4)$$

We are particularly interested in the term inside the summation in Eq.(4)

$$[s(yf_\theta(x)) - \lambda]_+ - [s(yf_\theta(x)) - \hat{\lambda}]_+ = \begin{cases} \hat{\lambda} - \lambda & s(yf_\theta(x)) > \hat{\lambda} \\ s(yf_\theta(x)) - \lambda & \lambda < s(yf_\theta(x)) \leq \hat{\lambda} \\ 0 & s(yf_\theta(x)) \leq \lambda \end{cases}.$$

According to this, at the level of individual training samples, the equivalent effect of using the `AoRR` loss is to uniformly reduce the individual losses by $\lambda$, but truncate the reduced individual loss at values below zero or above $\hat{\lambda}$. The situation is illustrated in Fig.2 for the logistic individual loss $s(yf(x)) = \log_2(1 + e^{-yf(x)})$, which is a convex and smooth surrogate to the ideal 01-loss. The effect of reducing and truncating from below and above has two interesting consequences. First, note that the use of convex and smooth surrogate loss inevitably introduce penalties to samples that are correctly classified but are "too close" to the boundary. The reduction of the individual loss alleviate that improper penalty. This property is also shared by the $\text{AT}_k$ loss. On the other hand, the ideal 01-loss exerts the same penalty to all incorrect classified samples regardless of their margin value, while the surrogate has unbounded penalties. This is the exact cause of the sensitivity to outliers of the previous aggregate losses, but the truncation of `AoRR` loss is similar to the 01-loss, and thus is more robust to the outliers. It is worth emphasizing that the above explanation of the `AoRR` loss has been illustrated at the individual sample level with fixed $\lambda$ and $\hat{\lambda}$. The aggregate `AoRR` loss defined by (4) as a whole is not an average sample-based loss because it can not be decomposed into the summation of individual losses over samples.

## 3.1 Classification Calibration

A fundamental question in learning theory for classification [2, 36] is to investigate when the best possible estimator from a learning objective is consistent with the best possible, *i.e.*, the Bayes rule. Here we investigate this statistical question for the AoRR loss by considering its infinite sample case, *i.e.*, $n \to \infty$. As mentioned above, the AoRR loss as a whole is not the average of individual losses over samples, and therefore the analysis for the standard ERM [2, 20] does not apply to our case.

We assume that the training data $\{(x_i, y_i)\}_{i=1}^n$ are i.i.d. from an unknown distribution $p$ on $\mathcal{X} \times \{\pm 1\}$. The misclassification error measures the quality of a classifier $f : \mathcal{X} \to \{\pm 1\}$ is denoted by $\mathcal{R}(f) = \Pr(Y \neq f(X)) = \mathbb{E}[\mathbb{I}_{Yf(X) \leq 0}]$. The Bayes error leads to the least expected error, which is defined by $\mathcal{R}^* = \inf_f \mathcal{R}(f) = f_c(x) = \text{sign}(\eta(x) - \frac{1}{2})$ where $\eta(x) = P(Y = 1 | X = x)$. It is well noted that, in practice, one uses a surrogate loss $\ell : \mathbb{R} \to [0, \infty)$ which is a continuous function and upper-bounds the 01-loss. Its true risk is given by $\mathcal{E}_\ell(f) = \mathbb{E}[\ell(Yf(X))]$. Denote the optimal $\ell$-risk by $\mathcal{E}_\ell^* = \inf_f \mathcal{E}_\ell(f)$, the *classification calibration* (point-wise form of Fisher consistency) for loss $\ell$ [2, 20] holds true if the minimizer $f_\ell^* = \inf_f \mathcal{E}_\ell(f)$ has the same sign as the Bayes rule $f_c(x)$, *i.e.*, $\text{sign}(f_\ell^*(x)) = \text{sign}(f_c(x))$ whenever $f_c(x) \neq 0$.

In analogy, we can investigate the classification calibration property of the AoRR loss. Specifically, we first obtain the population form of the AoRR loss using the infinite limit of the empirical one given by Eq.(4). Indeed, we know from [4, 6] that, for any bounded $f$ and $\alpha \in (0, 1]$, there holds $\inf_{\lambda \geq 0} \alpha\lambda + \frac{1}{n}\sum_{i=1}^n [s(y_i f_\theta(x_i)) - \lambda]_+ \to \inf_{\lambda \geq 0} \alpha\lambda + \mathbb{E}[s(Yf(X)) - \lambda]_+$ as $n \to \infty$. Consequently, we have the limit case of the AoRR loss $\mathcal{L}_{aorr}(S(\theta))$ restated as follows:

$$\frac{n}{k-m}\left[\min_\lambda\left\{\frac{k}{n}\lambda + \frac{1}{n}\sum_{i=1}^n[s(y_i f_\theta(x_i)) - \lambda]_+\right\} - \min_{\hat{\lambda}}\left\{\frac{m}{n}\hat{\lambda} + \frac{1}{n}\sum_{i=1}^n[s(y_i f_\theta(x_i)) - \hat{\lambda}]_+\right\}\right]$$

$$\xrightarrow[n\to\infty]{\frac{k}{n}\to\nu, \frac{m}{n}\to\mu} \frac{n}{k-m}\left[\min_{\lambda\geq 0}\left\{\mathbb{E}[[s(Yf(X)) - \lambda]_+] + \nu\lambda\right\} - \min_{\hat{\lambda}\geq 0}\left\{\mathbb{E}[[s(Yf(X)) - \hat{\lambda}]_+] + \mu\hat{\lambda}\right\}\right]. \quad (5)$$

Throughout the paper, we assume that $\nu > \mu$ which is reasonable as $k > m$. In particular, we assume that $\mu > 0$ since if $\mu = 0$ then it will lead to $\hat{\lambda} = \infty$ and this case is reduced to the population version of average top-k case in [11]. As such, the population version of our AoRR loss (4) is given by

$$(f_0^*, \lambda^*, \hat{\lambda}^*) = \arg\inf_{f, \lambda\geq 0}\sup_{\hat{\lambda}\geq 0}\left\{\mathbb{E}[[s(Yf(X)) - \lambda]_+ - [s(Yf(X)) - \hat{\lambda}]_+] + (\nu\lambda - \mu\hat{\lambda})\right\}. \quad (6)$$

It is difficult to directly work on the optima $f_0^*$ since the problem in Eq.(6) is a non-convex min-max problem and the standard min-max theorem does not apply here. Instead, we assume the existence of $\lambda^*$ and $\hat{\lambda}^*$ in (6) and work with the minimizer $f^* = \arg\inf_f \mathcal{L}(f, \lambda^*, \hat{\lambda}^*)$ where $\mathcal{L}(f, \lambda^*, \hat{\lambda}^*) := \mathbb{E}[[s(Yf(X)) - \lambda^*]_+ - [s(Yf(X)) - \hat{\lambda}^*]_+] + (\nu\lambda^* - \mu\hat{\lambda}^*)$. Now we can define the classification calibration for the AoRR loss.

**Definition 1** *The AoRR loss is called classification calibrated if there is a minimizer $f^* = \arg\inf_f \mathcal{L}(f, \lambda^*, \hat{\lambda}^*)$ such as $f^*(x) > 0$ if $\eta(x) > 1/2$ and $f^*(x) < 0$ if $\eta(x) < 1/2$.*

We can then obtain the following theorem. Its proof can be found in the Appendix A.3.

**Theorem 2** *Suppose the individual loss $s : \mathbb{R} \to \mathbb{R}^+$ is non-increasing, convex, differentiable at 0 and $s'(0) < 0$. If $0 \leq \lambda^* < \hat{\lambda}^*$, then the AoRR loss is classification calibrated.*

## 3.2 Experiments

We empirically demonstrate the effectiveness of the AoRR aggregate loss combined with two types of individual losses for binary classification, namely, the logistic loss and the hinge loss. For simplicity, we consider a linear prediction function $f(x; \theta) = \theta^T x$ with parameter $\theta$, and the $\ell_2$ regularizer $\frac{1}{2C}\|\theta\|_2^2$ with $C > 0$.

**Synthetic data.** We generate two sets of 2D synthetic data (Fig.3). Each dataset contains 200 samples from Gaussian distributions with different means and variances. We consider both the case of the balanced (Fig.3 (a,b)) and the imbalanced (Fig.3 (c,d)) data distributions, in the former the training data for the two classes are approximately equal while in the latter one class has a dominating number

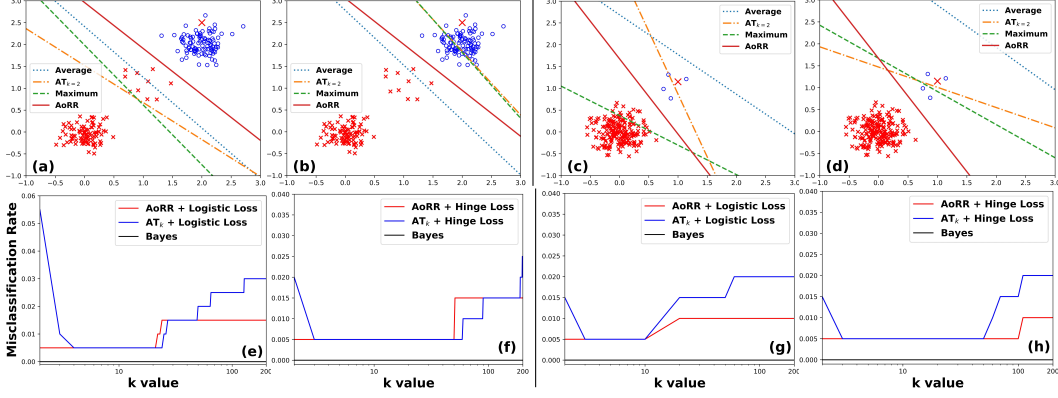

Figure 3: *Comparison of different aggregate losses for binary classification on a balanced but multi-modal synthetic dataset and with outliers with logistic loss (a) and hinge loss (b), and an imbalanced synthetic dataset with outliers with logistic loss (c) and hinge loss (d). Outliers in data are shown as × in blue class. The figures (e), (f), (g) and (h) show the misclassification rates of* `AoRR` *w.r.t. different values of k for each case and compare with the* $AT_k$ *and the optimal Bayes classifier.*

of samples in comparison to the other. The learned linear classifiers with different aggregate losses are shown in Fig.3. Both datasets have an outlier in the blue class (shown as ×). Experiments with more outliers can be found in the Appendix C.1.

To optimally remove the effect of outliers, we need to set $k$ larger than the number of outliers in the training dataset. Since there is one outlier in this synthetic dataset, we select $k = 2$ here as an example. As shown in Fig.3, neither the maximum loss nor the average loss performs well on the synthetic dataset, due to the existence of outliers and the multi-modal nature of the data. Furthermore, Fig.3 also shows that the $AT_k$ loss does not bode well: it is still affected by outliers. The reason can be that the training process with the $AT_k$ loss with $k = 2$ will most likely pick up one individual loss from the outlier for optimization. In contrast, the `AoRR` loss with $k$=2 and $m$=1, which is equivalent to the top-2 or second largest individual loss, yields better classification results. Intuitively, we avoid the direct effects of the outlier since it has the largest individual loss value. Furthermore, we perform experiments to show misclassification rates of `AoRR` with respect to different values of $k$ in Fig.3 (e), (f), (g), (h) for each case and compare with the $AT_k$ loss and optimal Bayes classifier. The results show that for $k$ values other than 2, the `AoRR` loss still exhibits an advantage over the $AT_k$ loss. Our experiments are based on a grid search for selecting the value of $k$ and $m$ because we found it is simple and often yields comparable performance. In practice for large-scale datasets, we can decide the minimal value of $m$ if we have prior knowledge about the faction of outliers in the dataset. To avoid extra freedom due to the value of $k$, we follow a very popular adaptive setting which has been applied in previous works, (e.g., [15]). At the beginning of training, $k$ equals to the size ($n$) of training data, $k = \lfloor \frac{n}{2} \rfloor$ once training accuracy $\geq 70\%$, $k = \lfloor \frac{n}{4} \rfloor$ once training accuracy $\geq 80\%$, $k = \lfloor \frac{n}{8} \rfloor$ once training accuracy $\geq 90\%$, $k = \lfloor \frac{n}{16} \rfloor$ once training accuracy $\geq 95\%$, $k = \lfloor \frac{n}{32} \rfloor$ once training accuracy $\geq 99.5\%$.

**Real data.** We use five benchmark datasets from the UCI [10] and the KEEL [1] data repositories (Statistical information of each dataset is given in the Appendix B.5). For each dataset, we first randomly select $50\%$ samples for training, and the remaining $50\%$ samples are randomly split for validation and testing (each contains $25\%$ samples). Hyper-parameters $C$, $k$, and $m$ are selected based on the validation set. Specifically, parameter $C$ is chosen from $\{10^0, 10^1, 10^2, 10^3, 10^4, 10^5\}$, parameter $k \in \{1\} \cup [0.1 : 0.1 : 1]n$, where $n$ is the number of training samples, and parameter $m$ are selected in the range of $[1, k)$. The following results are based on the optimal values of $k$ and $m$ obtained based on the validation set. The random splitting of the training/validation/testing sets is repeated 10 times and the average error rates, as well as the standard derivation on the testing set are reported in Table 1. In [30], the authors introduce slack variables to indicate outliers and propose a robust version of the maximum loss. We term it as Robust_Max loss and compare it to our method as one of the baselines. As these results show, comparing to the maximum, Robust_Max, average, and $AT_k$ losses, the `AoRR` loss achieves the best performance on all five datasets with both individual logistic loss and individual hinge loss. For individual logistic loss, the `AoRR` loss significantly improves the classification performance on `Monk` and `Phoneme` datasets and a slight improvement on datasets `Titanic` and `Splice`. More specifically, the performance of maximum aggregate loss

| Datasets | Logistic Loss | | | | | Hinge Loss | | | | |
|---|---|---|---|---|---|---|---|---|---|---|
| | Maximum | R_Max | Average | $AT_k$ | AoRR | Maximum | R_Max | Average | $AT_k$ | AoRR |
| Monk | 22.41 | 21.69 | 20.46 | 16.76 | **12.69** | 22.04 | 20.61 | 18.61 | 17.04 | **13.17** |
| | (2.95) | (2.62) | (2.02) | (2.29) | **(2.34)** | (3.08) | (3.38) | (3.16) | (2.77) | **(2.13)** |
| Australian | 19.88 | 17.65 | 14.27 | 11.7 | **11.42** | 19.82 | 15.88 | 14.74 | 12.51 | **12.5** |
| | (6.64) | (1.3) | (3.22) | (2.82) | **(1.01)** | (6.56) | (1.05) | (3.10) | (4.03) | **(1.55)** |
| Phoneme | 28.67 | 26.71 | 25.50 | 24.17 | **21.95** | 28.81 | 24.21 | 22.88 | 22.88 | **21.95** |
| | (0.58) | (1.4) | (0.88) | (0.89) | **(0.71)** | (0.62) | (1.7) | (1.01) | (1.01) | **(0.68)** |
| Titanic | 26.50 | 24.15 | 22.77 | 22.44 | **21.69** | 25.45 | 25.08 | 22.82 | 22.02 | **21.63** |
| | (3.35) | (3.12) | (0.82) | (0.84) | **(0.99)** | (2.52) | (1.2) | (0.74) | (0.77) | **(1.05)** |
| Splice | 23.57 | 23.48 | 17.25 | 16.12 | **15.59** | 23.40 | 22.82 | 16.25 | 16.23 | **15.64** |
| | (1.93) | (0.76) | (0.93) | (0.97) | **(0.9)** | (2.10) | (2.63) | (1.12) | (0.97) | **(0.89)** |

Table 1: *Average error rate (%) and standard derivation of different aggregate losses combined with individual logistic loss and hinge loss over 5 datasets. The best results are shown in bold. (R_Max: Robust_Max)*

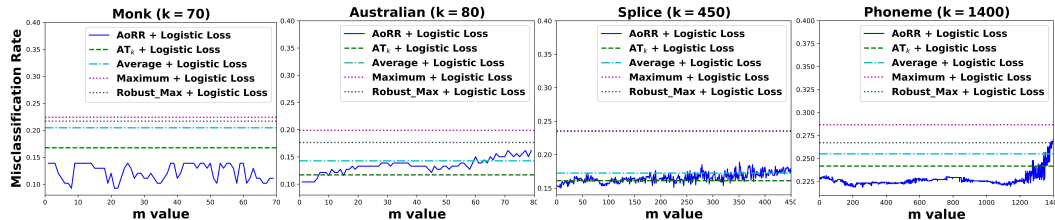

Figure 4: *Tendency curves of error rate of learning AoRR loss w.r.t. $m$ on four datasets.*

is very poor on all cases due to its high sensitivity to the outliers or noisy data. The optimization of the Robust_Max loss uses convex relaxation on the domain of slack variables constraint and using $l_2$ norm to replace the $l_1$ norm in the constraint. Therefore, it can alleviate the sensitivity to outliers, but cannot exclude the influence of them. The average aggregate loss is more robust to noise and outliers than the maximum loss and the Robust_Max loss on all datasets. However, as data distributions may be very complicated, the average loss may sacrifice samples from rare distributions to pursue a lower loss on the whole training set and obtains sub-optimal solutions accordingly. The $AT_k$ loss is not completely free from the influence of outliers and noisy data either, which can be observed in particular on the Monk dataset. On the Monk dataset, in comparison to the $AT_k$ loss, the AoRR loss reduce the misclassification rates by $4.07\%$ for the individual logistic loss and $3.87\%$ for the individual hinge loss, respectively.

To further compare with the $AT_k$ loss, we investigate the influence of $m$ in the AoRR loss. Specifically, we select the best $k$ value based on the $AT_k$ results, and vary $m$ in the range of $[1, k-1]$. We use the individual logistic loss and plot tendency curves of misclassification error rates w.r.t $m$ in Fig. 4, together with those from the average, maximum and Robust_Max losses. As these plots show, on all four datasets, there is a clear range of $m$ with better performance than the corresponding $AT_k$ loss. We observe a trend of decreasing error rates with $m$ increasing. This is because outliers correspond to large individual losses, and excluding them from the training loss helps improve the overall performance of the learned classifier. However, when $m$ becomes large, the classification performance is decreasing, as many samples with small losses are included in the AoRR objective and dominate the training process. The results for the individual hinge loss can be found in the Appendix C.2.

## 4 Multi-label Learning

We use SoRR to construct the individual loss for multi-label/multi-class classification, where a sample $x$ can be associated with a set of labels $\emptyset \neq Y \subset \{1, \cdots, l\}$. Our goal is to construct a linear predictor $f_\Theta(x) = \Theta^T x$ with $\Theta = (\theta_1, \cdots, \theta_l)$. The final classifier outputs labels for $x$ with the top $k$ ($1 \leq k < l$) prediction scores, *i.e.*, $\theta_{[1]}^\top x \geq \theta_{[2]}^\top x \geq \cdots \geq \theta_{[k]}^\top x$. In training, the classifier is expected to include as many true labels as possible in the top $k$ outputs. This can be evaluated by the "margin", *i.e.*, the difference between the $(k+1)$-th largest score of all the labels, $\theta_{[k+1]}^\top x$ and the lowest prediction score of all the ground-truth labels, $\min_{y \in Y} \theta_y^\top x$. If we have $\theta_{[k+1]}^\top x < \min_{y \in Y} \theta_y^\top x$, then all ground-truth labels have prediction scores ranked in the top $k$ positions. If this is not the case, then at least one ground-truth label has a prediction score not ranked in the top $k$. This induces the following metric for multi-label classification as $\mathbb{I}_{[\theta_{[k+1]}^\top x \geq \min_{y \in Y} \theta_y^\top x]}$. Replacing the indicator function with

the hinge function and let $S(\theta) = \{s_j(\theta)\}_{j=1}^l$, where $s_j(\theta) = \left[1 + \theta_j^\top x - \min_{y \in Y} \theta_y^\top x\right]_+$, we obtain a continuous surrogate loss, as $\psi_{k,k+1}(S(\theta)) = s_{[k+1]}(\theta)$. We term this loss as the *top-k multi-label* (TKML) loss. When $k = |Y|$, we have the following proposition and its proof can be found in the Appendix A.4,

**Proposition 1** *The* TKML *loss is a lower-bound to the conventional multi-label loss [8], as* $\left[1 + \max_{y \notin Y} \theta_y^\top x - \min_{y \in Y} \theta_y^\top x\right]_+ \geq \psi_{|Y|,|Y|+1}(S(\theta))$.

The TKML loss generalizes the conventional multi-class loss ($|Y| = k = 1$) and the top-$k$ consistent $k$-guesses multi-class classification [38] ($1 = |Y| \leq k < l$). A similar learning objective is proposed in [16] corresponds to $s_{[k]}(\theta)$, however, as proved in [38], it is not multi-class top-$k$ consistent. Another work in [7] proposes a robust top-k multi-class SVM based on the convex surrogate of $s_{[k]}(\theta)$ to address the outliers by using a hyperparameter to cap the values of the individual losses. This approach is different from ours since we directly address the original top-k multi-class SVM problem using our TKML loss without introducing its convex surrogate and it is consistent. For a set of training data $(x_1, Y_1), \cdots, (x_n, Y_n)$, if we denote $\psi_{k,k+1}(S(x, Y; \Theta)) = \psi_{k,k+1}(S(\theta))$, the data loss on TKML can be written as $\mathcal{L}_{TKML}(\Theta) = \frac{1}{n} \sum_{i=1}^n \psi_{k,k+1}(S(x_i, Y_i; \Theta))$, which can be optimized using the Algorithm 1.

## 4.1 Experiments

We use the same $\ell_2$ regularizer, $R(\theta) = \frac{1}{2C}||\theta||_2^2$ and cross-validate hyper-parameter $C$ in the range $10^0$ to $10^5$, extending it when the optimal value appears.

**Multi-label classification.** We use three benchmark datasets (Emotions, Scene, and Yeast) from the KEEL data repository to verify the effectiveness of our TKML loss. The average number of positive labels per instance in each dataset is 1.81, 1.06, and 4.22, respectively. For comparison, we compare TKML with logistic regression (LR) model (i.e. minimize a surrogate hamming loss [39]), and a ranking based method (LSEP [19]). For these two baseline methods, we use a sigmoid operator on the linear predictor as $f_\Theta(x) = 1/(1 + \exp(-\Theta^T x))$. Since TKML is based on the value of $k$, we use five different $k$ values ($k \in \{1, 2, 3, 4, 5\}$) to evaluate the performance. For each dataset, we randomly partition it to 50%/25%/25% samples for training/validation/testing, respectively. This random partition is repeated 10 times, and the average performance on testing data is reported in Table 2. We use a metric (top $k$ multi-label accuracy) $\frac{1}{n} \sum_{i=1}^n \mathbb{I}_{[(Z_i \subseteq Y_i) \vee (Y_i \subseteq Z_i)]}$ to evaluate the performance, where $n$ is the size of the sample set. For instance, $(x_i, Y_i)$ with $Y_i$ be its ground-truth set, $f_\Theta(x_i) \in \mathbb{R}^l$ be its predicted scores, and $Z_i$ be a set of top $k$ predictions according to $f_\Theta(x_i)$. This metric reflects the performance of a classifier can get as many true labels as possible in the top $k$ range. More details about these three datasets and settings can be found in the Appendix B.7 and B.8.

From Table 2, we note that the TKML loss in general improves the performance on the Emotions dataset for all different $k$ values. These results illustrate the effectiveness of the TKML loss. More specifically, our TKML method obtains 4.63% improvement on $k = 2$ and 4.42% improvement on $k = 3$ when comparing to LR. This rate of improvement becomes higher (6.26% improvement on $k = 2$) when compare to LSEP. We also compare the performance of the method based on the TKML loss on different $k$ values. If we choose the value of $k$ close to the number of the ground-truth labels, the corresponding classification method outperforms the two baseline methods. For example, in the case of the Emotions dataset, the average number of positive labels per instance is 1.81, and our method based on the TKML loss achieves the best performance for $k = 1, 2$. As another example, the average number of true labels for the Yeast dataset is 4.22, so the method based on the TKML loss achieves the best performance for $k = 4, 5$. We provide more experiments in Appendix C.3.

**Robustness analysis.** As a special case of AoRR, the TKML loss exhibit similar robustness with regards to outliers, which can be elucidated with experiments in the multi-class setting (*i.e.*, $k$=1 and $|Y|$=1). We use the MNIST dataset[18], which contains $60,000$ training samples and $10,000$ testing samples that are images of handwritten digits. To simulate outliers caused by errors occurred when labeling the data, as in the work of [37], we use the asymmetric (class-dependent) noise creation method [26, 40] to randomly change labels of the training data ($2 \rightarrow 7$, $3 \rightarrow 8$, $5 \leftrightarrow 6$, and $7 \rightarrow 1$) with a given proportion. The flipping label is chosen at random with probability $p = 0.2, 0.3, 0.4$. As a baseline, we use the top-$k$ multi-class SVM ($\text{SVM}_\alpha$) [16]. The performance is evaluated with the top 1, top 2, $\cdots$, top 5 accuracy on the testing samples.

| Datasets | Methods | $k=1$ | $k=2$ | $k=3$ | $k=4$ | $k=5$ |
|---|---|---|---|---|---|---|
| | LR | 73.54(3.98) | 57.48(3.35) | 73.20(4.69) | 86.60(3.02) | 96.46(1.71) |
| Emotions | LSEP | 72.18(4.56) | 55.85(3.37) | 72.18(3.74) | 85.58(2.92) | 95.85(1.07) |
| | TKML | **76.80(2.66)** | **62.11(2.85)** | **77.62(2.81)** | **90.14(2.22)** | **96.94(0.63)** |
| | LR | 73.2(0.57) | 85.31(0.47) | **94.79(0.79)** | **97.88(0.63)** | **99.7(0.30)** |
| Scene | LSEP | 69.22(3.43) | 83.83(4.83) | 92.46(4.78) | 96.35(3.5) | 98.56(1.94) |
| | TKML | **74.06(0.45)** | **85.36(0.79)** | 88.92(1.47) | 91.94(0.87) | 95.01(0.61) |
| | LR | **77.57(0.91)** | **70.59(1.16)** | **52.65(1.23)** | 43.26(1.16) | 43.49(1.33) |
| Yeast | LSEP | 75.5(1.03) | 66.84(2.9) | 49.72(1.26) | 41.90(1.91) | 43.01(1.02) |
| | TKML | 76.94(0.49) | 67.19(2.79) | 45.41(0.71) | **43.47(1.06)** | **44.69(1.14)** |

Table 2: *Top k multi-label accuracy with its standard derivation (%) on three datasets. The best performance is shown in bold.*

| Noise Level | Methods | Top-1 Accuracy | Top-2 Accuracy | Top-3 Accuracy | Top-4 Accuracy | Top-5 Accuracy |
|---|---|---|---|---|---|---|
| 0.2 | $SVM_\alpha$ | 78.33(0.18) | 90.66(0.29) | 95.12(0.2) | 97.28(0.09) | 98.49(0.1) |
| | TKML | **83.06(0.94)** | **94.17(0.19)** | **97.24(0.13)** | **98.47(0.05)** | **99.22(0.01)** |
| 0.3 | $SVM_\alpha$ | 74.65(0.17) | 89.31(0.24) | 94.14(0.2) | 96.73(0.23) | 98.19(0.07) |
| | TKML | **80.13(1.24)** | **93.37(0.1)** | **96.81(0.22)** | **98.21(0.05)** | **99.08(0.05)** |
| 0.4 | $SVM_\alpha$ | 68.32(0.32) | 86.71(0.42) | 93.14(0.49) | 96.16(0.32) | 97.84(0.18) |
| | TKML | **75(1.15)** | **92.41(0.14)** | **96.2(0.13)** | **97.95(0.1)** | **98.89(0.04)** |

Table 3: *Testing accuracy (%) of two methods on MNIST with different levels of asymmetric noisy labels. The average accuracy and standard deviation of 5 random runs are reported and the best results are shown in bold.*

From Table 3, it is clear that our method TKML consistently outperforms the baseline $SVM_\alpha$ among all top 1-5 accuracies. The gained improvement in performance is getting more significant as the level of noise increases. Since our flipping method only works between two different labels, we expected the performance of TKML has some significant improvements on top 1 and 2 accuracies. Indeed, this expectation is correctly verified as Table 3 clearly indicates that the performance of our method is better than $SVM_\alpha$ by nearly 7% accuracy (see Top-1 accuracy in the noise level 0.4). These results also demonstrate our optimization framework works well. More experiments can be found in the Appendix C.4.

# 5 Conclusion

In this work, we introduce a general approach to form learning objectives, *i.e.*, sum of ranked range, which corresponds to the sum of a consecutive sequence of sorted values of a set of real numbers. We show that SoRR can be expressed as the difference between two convex problems and optimized with the difference-of-convex algorithm (DCA).

We explore two applications in machine learning of the minimization of the SoRR framework, namely the AoRR aggregate loss for binary classification and the TKML individual loss for multi-label/multi-class classification. Our empirical results showed the effectiveness of the proposed framework on achieving superior generalization and robust performance on synthetic and real datasets. For future works, we plan to further study the consistency of TKML loss for multi-label learning and incorporate SoRR into the learning of deep neural networks.

# 6 Broader Impact

Loss functions are fundamental components in any machine learning system. Our work, by designing new types of loss functions based on the use of SoRR, is expected to be applicable to a wide range of ML problems. The benefit of using our method is the better handling of potential outliers in the training dataset, which could be the result of gross error or intentional "poisoning" of the dataset. However, there is also a risk of resulting a biased learning model when certain training samples are excluded. To mitigate such risks, we encourage further study to understand the impacts of using SoRR based losses in particular real-world scenarios, focusing on the more contextually meaning choice of the values $m$ and $k$ for better tradeoff of robustness and bias.

**Acknowledgments**. We are grateful to all anonymous reviewers for their constructive comments. This work is supported by NSF research grants (IIS-1816227 and IIS-2008532) as well as an Army Research Office grant (agreement number: W911 NF-18-1-0297).

## Footnotes

[1]Code available at `https://github.com/discovershu/SoRR`.

[2]For large datasets, we can use a stochastic version of DCA, which is more efficient with a provable convergence to a critical point [34].

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
