[Supplementary Material]

# Appendix

## A  Proofs

### A.1  Proof of Theorem 1

To prove Theorem 1, we need the following lemma.

**Lemma 1 ([24])** . $\phi_k(S)$ *is a convex function of the elements of S. Furthermore, for any* $i \in [1, n]$, *we have* $\sum_{i=1}^{k} s_{[i]} = min_{\lambda \in \mathbb{R}}\{k\lambda + \sum_{i=1}^{n}[s_i - \lambda]_+\}$, *of which* $s_{[k]}$ *is an optimum solution.*

**Proof of Theorem 1**

**Proof:** From Lemma 1, we have

$$\min_{\theta} \psi_{m,k}(S(\theta)) = \min_{\theta}\big[\phi_k(S(\theta)) - \phi_m(S(\theta))\big]$$

$$= \min_{\theta}\left[\min_{\lambda \in \mathbb{R}}\left\{k\lambda + \sum_{i=1}^{n}[s_i(\theta) - \lambda]_+\right\} - \min_{\hat{\lambda} \in \mathbb{R}}\left\{m\hat{\lambda} + \sum_{i=1}^{n}[s_i(\theta) - \hat{\lambda}]_+\right\}\right].$$

If the optimal solution $\theta^*$ is achieved, from Lemma 1, we get $\lambda = s_{[k]}$ and $\hat{\lambda} = s_{[m]}$. Therefore, $\hat{\lambda} > \lambda$ because $k > m$. ∎

### A.2  Proof of Equation (3)

Before introducing the sub-gradient of $\phi_m(S(\theta))$, we provide a very useful characterization of differentiable properties of the optimal value function [3, Proposition A.22], which is also an extension of Danskin's theorem [9].

**Lemma 2** *Let* $\phi : \mathbb{R}^n \times \mathbb{R}^m \rightarrow (-\infty, \infty]$ *be a function and let Y be a compact subset of* $\mathbb{R}^m$. *Assume further that for every vector* $y \in Y$ *the function* $\phi(\cdot, y) : \mathbb{R}^n \rightarrow (-\infty, \infty]$ *is a closed proper convex function. Consider the function f defined as* $f(x) = sup_{y \in Y}\phi(x, y)$, *then if f is finite somewhere, it is a closed proper convex function. Furthermore, if* $int(dom f) \neq \emptyset$ *and* $\phi$ *is continuous on the set* $int(dom f) \times Y$, *then for every* $x \in int(dom f)$ *we have* $\partial f(x) = conv\{\partial\phi(x, \overline{y})|\overline{y} \in \overline{Y}(x)\}$, *where* $\overline{Y}(x)$ *is the set* $\overline{Y}(x) = \{\overline{y} \in Y|\phi(x, \overline{y}) = max_{y \in Y}\phi(x, y)\}$

**Proof of Equation (3)**

**Proof:** We apply Lemma 2 with a new notation $\phi_m(\theta, \hat{\lambda}) = m\hat{\lambda} + \sum_{i=1}^{n}[s_i(\theta) - \hat{\lambda}]_+$. Suppose $\theta \in \mathbb{R}^n$ and $\hat{\lambda} \in \mathbb{R}$, the function $\phi_m : \mathbb{R}^n \times \mathbb{R} \rightarrow (-\infty, \infty]$. Let Y be a compact subset of $\mathbb{R}$ and for every $\hat{\lambda} \in Y$, it is obviously that the function $\phi_m(\cdot, \hat{\lambda}) : \mathbb{R}^n \rightarrow (-\infty, \infty]$ is a closed proper convex function w.r.t $\theta$ from the second term of Eq.(1).

Consider a function $f$ defined as $f(\theta) = \sup_{\hat{\lambda} \in Y}\phi(\theta, \lambda)$, since $f$ is finite somewhere, it is a closed proper convex function. The interior of the effective domain of $f$ is nonempty, and that $\phi_m$ is continuous on the set $int(dom f) \times Y$. The condition of lemma 2 is satisfied.

$\forall \theta \in int(dom f)$, we have

$$\partial f(\theta) = conv\{\partial\phi_m(\theta, \overline{\lambda})|\overline{\lambda} \in \overline{Y}(\theta)\},$$

where

$$\overline{Y}(\theta) = \{\overline{\lambda} \in Y|\phi_m(\theta, \overline{\lambda}) = max_{\hat{\lambda} \in Y}\phi_m(\theta, \hat{\lambda})\} = \{\overline{\lambda} \in Y| - \phi_m(\theta, \overline{\lambda}) = -min_{\hat{\lambda} \in Y}\phi_m(\theta, \hat{\lambda})\}.$$

As we know $-min_{\hat{\lambda} \in Y}\phi_m(\theta, \hat{\lambda}) = -\phi_m(S(\theta))$. This means the subdifferential of $f$ w.r.t $\theta$ exists when we set the optimal value of $\hat{\lambda}$.

From the above and the lemma 1, we can get the sub-gradient $\hat{\theta} \in \partial\phi_m(S(\theta)) = \sum_{i=1}^{n} \partial s_i(\theta) \cdot \mathbb{I}_{[s_i(\theta) > \hat{\lambda}]}$, where $\hat{\lambda}$ equals to $s_{[m]}(\theta)$.

∎

## A.3 Proof of Theorem 2

**Proof:** Without loss of generality, by normalization we can assume $s(0) = 1$ which can be satisfied by scaling. For any fixed $x \in \mathcal{X}$, by the definition of $f^* = \arg \inf \mathcal{L}(f, \lambda^*, \hat{\lambda}^*)$, we know that

$$f^*(x) = t^* = \arg \inf_{t \in \mathbb{R}} \mathbb{E}\Big[ [s(Yt) - \lambda^*]_+ - [s(Yt) - \hat{\lambda}^*]_+ \Big| X = x \Big].$$

Notice the assumption $\hat{\lambda}^* > \lambda^*$ and recall $\eta(x) = P(y = 1|x)$. We need to show that $t^* > 0$ for $\eta(x) > 1/2$ and $t^* < 0$ if $\eta(x) < 1/2$. Indeed, if $t^* \neq 0$, then, by the definition of $t^*$, we have that

$$\mathbb{E}\Big[ [s(Yt^*) - \lambda^*]_+ - [s(Yt^*) - \hat{\lambda}^*]_+ \Big| X = x \Big] < \mathbb{E}\Big[ [s(-Yt^*) - \lambda^*]_+ - [s(-Yt^*) - \hat{\lambda}^*]_+ \Big| X = x \Big]$$

The above inequality is identical to

$$\Big[ \big( (s(t^*) - \lambda^*)_+ - (s(t^*) - \hat{\lambda}^*)_+ \big) - \big( (s(-t^*) - \lambda^*)_+ - (s(-t^*) - \hat{\lambda}^*)_+ \big) \Big] \big[ 2\eta(x) - 1 \big] < 0.$$

Since $\hat{\lambda}^* > \lambda^*$, we have that that $g(s) = (s - \lambda^*)_+ - (s - \hat{\lambda}^*)_+$ is a non-decreasing function of variable $s$. Then, if $\eta(x) > \frac{1}{2}$ we must have $g(s(t^*)) < g(s(-t^*))$ which indicates $s(t^*) < s(-t^*)$. From the non-increasing property of $s$ on $\mathbb{R}$, $s(t)$ is also a convex function and $s'(0) < 0$ immediately indicates $t^* > 0$. Likewise, we can show that $t^* < 0$ for $\eta(x) < 1/2$.

To prove $t = 0$ is not a minimizer, without loss of generality, assume $\eta(x) > \frac{1}{2}$. We need to consider two conditions as follows,

1. If $0 \leq \lambda^* < \hat{\lambda}^* \leq 1$ and $s(0) = 1$, then

$$A = \mathbb{E}\Big[ [s(0) - \lambda^*]_+ - [s(0) - \hat{\lambda}^*]_+ \Big| X = x \Big]$$
$$= [1 - \lambda^*]_+ - [1 - \hat{\lambda}^*]_+$$
$$= \hat{\lambda}^* - \lambda^*$$

Since $s'(0) < 0$ and $s$ is non-increasing, there exists $t^0 > t^* = 0 > -t^0$, and $s(-t^0) > s(0) \geq \hat{\lambda}^* > s(t^0) > \lambda^*$. Let

$$B = \mathbb{E}\Big[ [s(Yt^0) - \lambda^*]_+ - [s(Yt^0) - \hat{\lambda}^*]_+ | X = x \Big]$$
$$= \Big( [s(t^0) - \lambda^*]_+ - [s(t^0) - \hat{\lambda}^*]_+ \Big) \eta(x) + \Big( [s(-t^0) - \lambda^*]_+ - [s(-t^0) - \hat{\lambda}^*]_+ \Big) \Big( 1 - \eta(x) \Big)$$
$$= \Big( [s(-t^0) - \lambda^*]_+ - [s(-t^0) - \hat{\lambda}^*]_+ \Big)$$
$$\quad + \Big[ \Big( [s(t^0) - \lambda^*]_+ - [s(t^0) - \hat{\lambda}^*]_+ \Big) - \Big( [s(-t^0) - \lambda^*]_+ - [s(-t^0) - \hat{\lambda}^*]_+ \Big) \Big] \eta(x)$$
$$= \hat{\lambda}^* - \lambda^* + \Big[ s(t^0) - \lambda^* - (\hat{\lambda}^* - \lambda^*) \Big] \eta(x)$$

Then

$$B - A = (s(t^0) - \hat{\lambda}^*) \eta(x) < 0$$

Therefore, $t = 0$ is not a minimizer.

2. If $0 \leq \lambda^* \leq 1 < \hat{\lambda}^*$ and $s(0) = 1$, then

$$\frac{d}{dt} \mathbb{E}[[s(Yt) - \lambda^*]_+ - [s(Yt) - \hat{\lambda}^*]_+]|_{t=0}$$
$$= \frac{d}{dt} [\eta(x)([s(t) - \lambda^*]_+ - [s(t) - \hat{\lambda}^*]_+) + (1 - \eta(x))([s(-t) - \lambda^*]_+ - [s(-t) - \hat{\lambda}^*]_+)]|_{t=0}$$
$$= \frac{d}{dt} [\eta(x)(s(t) - \lambda^*) + (1 - \eta(x))(s(-t) - \lambda^*)]|_{t=0}$$
$$= [\eta(x)s'(t) - (1 - \eta(x))s'(-t)]|_{t=0}$$
$$= (2\eta(x) - 1)s'(0) < 0$$

Thus $t = 0$ is not a minimizer. ∎

## A.4 Proof of Proposition 1

**Proof:** We just need to prove that $\max_{y \notin Y} \theta_y^\top x \geq \theta_{[|Y|+1]}^\top x$. If this is not the case, then for any label $y \notin Y$, then its rank in the ranked list is no more than $|Y| + 2$, then the sum of total number of such labels is not larger than $l - (|Y| + 2) + 1 = l - |Y| - 1$. And the total number of labels will be $|Y| + |\{y \notin Y\}| \leq l - 1 \neq l$, which is a contradiction. ∎

# B Additional Experimental Details

## B.1 Source Code

For the purpose of review, the source code and datasets are accessible at supplementary file.

## B.2 Computing Infrastructure Description

All algorithms are implemented in Python 3.6 and trained and tested on an Intel(R) Xeon(R) CPU W5590 @3.33GHz with 48GB of RAM.

## B.3 Time Complexity Analyze

We consider the average case in the time complexity analyze. For a given outer loop size $|t|$, a inner loop size $|l|$, and training sample size $n$, the complexity of our MSoRR Algorithm 1 is $O(|t|(n \log n + |l|))$.

## B.4 Training Settings on Toy Examples for Aggregate Loss

To reproduce the experimental results of `AoRR` on synthetic data, we provide the details about the settings when we are training the model in Table 4. For example, the learning rate, the number of epochs for the outer loop, and the number of epochs for the inner loop.

| Datasets | Outliers | Logistic loss | | | Hinge loss | | |
|---|---|---|---|---|---|---|---|
| | | LR | # OE | # IE | LR | # OE | # IE |
| Multi-modal data | 1 | 0.01 | 100 | 1000 | 0.01 | 5 | 1000 |
| | 2 | 0.01 | 100 | 1000 | 0.01 | 5 | 1000 |
| | 3 | 0.01 | 100 | 1000 | 0.01 | 5 | 1000 |
| | 4 | 0.01 | 100 | 1000 | 0.01 | 5 | 1000 |
| | 5 | 0.01 | 100 | 1000 | 0.01 | 5 | 1000 |
| | 10 | 0.01 | 100 | 1000 | 0.01 | 5 | 1000 |
| | 20 | 0.01 | 100 | 1000 | 0.01 | 5 | 1000 |
| Imbalanced data | 1 | 0.01 | 100 | 1000 | 0.01 | 5 | 1000 |

LR: Learning Rate, OE: Outer Epochs, IE: Inner epochs

Table 4: *AoRR settings on toy experiments.*

## B.5 Description of Datasets for Aggregate Loss

In aggregate loss experiments, for real-world datasets, we use five benchmark datasets from the UCI and the KEEL data repositories. The details of these datasets are shown in Table 5.

| Datasets | #Classes | #Samples | #Features | Class Ratio |
|---|---|---|---|---|
| Monk | 2 | 432 | 6 | 1.12 |
| Australian | 2 | 690 | 14 | 1.25 |
| Phoneme | 2 | 5,404 | 5 | 2.41 |
| Titanic | 2 | 2,201 | 3 | 2.10 |
| Splice | 2 | 3,175 | 60 | 1.08 |

Table 5: *Statistical information of each dataset for aggregate loss.*

## B.6 Training Settings on Real Datasets for Aggregate Loss

We provide a reference for setting parameters to reproduce our `AoRR` experiments on real datasets. Table 6 contains the settings for individual logistic loss. Table 7 is for individual hinge loss.

| Datasets | $k$ | $m$ | $C$ | # Outer epochs | # Inner epochs | Learning rate |
|---|---|---|---|---|---|---|
| Monk | 70 | 20 | $10^4$ | 5 | 2000 | 0.01 |
| Australian | 80 | 3 | $10^4$ | 10 | 1000 | 0.01 |
| Phoneme | 1400 | 100 | $10^4$ | 10 | 1000 | 0.01 |
| Titanic | 500 | 10 | $10^4$ | 10 | 1000 | 0.01 |
| Splice | 450 | 50 | $10^4$ | 10 | 1000 | 0.01 |

Table 6: *AoRR settings on real datasets for individual logistic loss.*

| Datasets | $k$ | $m$ | $C$ | # Outer epochs | # Inner epochs | Learning rate |
|---|---|---|---|---|---|---|
| Monk | 70 | 45 | $10^4$ | 5 | 1000 | 0.01 |
| Australian | 80 | 3 | $10^4$ | 5 | 1000 | 0.01 |
| Phoneme | 1400 | 410 | $10^4$ | 10 | 500 | 0.01 |
| Titanic | 500 | 10 | $10^4$ | 5 | 500 | 0.01 |
| Splice | 450 | 50 | $10^4$ | 10 | 1000 | 0.01 |

Table 7: *AoRR settings on real datasets for individual hinge loss.*

## B.7 Description of Datasets for Multi-label Learning

In multi-label learning experiments, we conduct experiments on three benchmark datasets (Emotions, Scene and Yeast) from the KEEL data repository. The details of them as described in Table 8.

| Datasets | #Samples | #Features | #Labels | $\bar{c}$ |
|---|---|---|---|---|
| Emotions | 593 | 72 | 6 | 1.81 |
| Scene | 2,407 | 294 | 6 | 1.06 |
| Yeast | 2,417 | 103 | 14 | 4.22 |

Table 8: *Statistical information of each dataset for multi-label learning, where $\bar{c}$ represents the average number of positive labels per instance.*

## B.8 Training Settings for Multi-label Learning

The settings for TKML on three real datasets are shown in Table 9.

| Datasets | $C$ | #Outer epochs | #Inner epochs | Learning rate |
|---|---|---|---|---|
| Emotions | $10^4$ | 20 | 1000 | 0.1 |
| Scene | $10^4$ | 20 | 1000 | 0.1 |
| Yeast | $10^4$ | 20 | 1000 | 0.1 |

Table 9: *TKML settings on each dataset.*

## B.9 Training Settings for Multi-class Learning

Training settings for the MNIST dataset in different noise level can be found in Table 10.

| Noise level | #Outer epochs | #Inner epochs | Learning rate |
|---|---|---|---|
| 0.2 | 27 | 2000 | 0.1 |
| 0.3 | 25 | 2000 | 0.1 |
| 0.4 | 21 | 2000 | 0.1 |

Table 10: *TKML settings on the MNIST dataset in different noise levels.*

# C  Additional Experimental Results

## C.1  Toy Examples with More Outliers for Effects of Aggregate Losses

In order to evaluate the effects of different aggregate losses on more than one outlier, we also conducted additional experiments on a multi-modal toy example with outliers. We use Gaussian distributions with the different mean and standard deviations to generate this dataset (Fig.5). It contains 200 samples and is distributed in 2 classes (100 samples in red class and 100 samples in blue class). The red samples are sampled from two distributions (primary distribution and minor distribution). The blue samples are sampled from only one distribution. However, they can still be separated. A linear classifier is considered and different aggregate losses are evaluated in individual logistic loss (i.e., Fig.5 (a), (c), (e), (g), (i), (k)) and individual hinge loss (i.e., Fig.5 (b), (d), (f), (h), (j), (l)). Given a number $n$, we set outliers as replacing $n$ blue samples class with the opposite class. The outliers have been shown as $\times$ in blue class. For $AT_k$ and AoRR losses, we let the value of $k$ be the same and equals to $n + 1$. Let the value of $m$ equals to $n$ in AoRR loss. We consider six cases as follows,

**Case 1 (2 outliers).** In Fig.5 (a) and (b), there exist two outliers. Let hyper-parameters $k = 3$ and $m = 2$.

**Case 2 (3 outliers).** Fig.5 (c) and (d) contain three outliers. In this scenario, $k = 4$ and $m = 3$.

**Case 3 (4 outliers).** Fig.5 (e) and (f) include four outliers and we set $k = 5$ and $m = 4$.

Figure 5: *Comparison of different aggregate losses on 2D synthetic data with 200 samples for binary classification with individual logistic loss (a, c, e, g, i, k) and individual hinge loss (b, d, f, h, j, l). Outliers are shown as × in blue class.*

**Case 4 (5 outliers).** There are five outliers in Fig.5 (g) and (h). We set $k = 6$ and $m = 5$.

**Case 5 (10 outliers).** Ten outliers have been included in Fig.5 (i) and (j). Let $k = 11$ and $m = 10$ in this case.

**Case 6 (20 outliers).** We create twenty outliers in Fig.5 (k) and (l) and make $k = 21$ and $m = 20$.

See from case 1, 2, 3, 4, the linear classifier learned from average aggregate loss cross some red samples from minor distribution even though the data is separable. The reason is that the samples close to the decision boundary are sacrificed to reduce the total loss over the whole dataset.

Since the $k$ value is set to be $n + 1$, the $\text{AT}_k$ loss select $k$ largest individual losses which contain many outliers to train the classifier. It leads to the instability of the learned classifier. This phenomenon can be found when we compare all cases. Similarly, the maximum aggregate loss cannot fit this data very well in all cases. This loss is very sensitive to outliers.

From cases 5 and 6, the average aggregate loss with individual logistic loss achieves better results than with individual hinge loss. A possible reason is that for correctly classified samples with a margin greater than 1, the penalty caused by hinge loss is 0. However, it is non-zero when using logistic loss. Since many outliers in blue class, to reduce the average loss, the decision boundary will close to blue class. Especially, when we compare (i) and (k), it is obvious that average loss can achieve a better result while the number of outliers is increasing.

As we discussed, hinge loss has less penalty for correctly classified samples than logistic loss. This causes outliers to be more prominent than normal samples while using the individual hinge loss. This analysis can be verified in the experiment when we compare the individual logistic loss and the individual hinge loss. For example, (i) and (j), (k) and (l), etc.. We find the decision boundaries of maximum loss and $\text{AT}_k$ loss are close to outliers in the individual hinge loss scenario because both of them are sensitive to outliers in our cases.

## C.2 Additional Tendency Curves for Effects of Aggregate Losses

In this section, we use individual hinge loss as an example and plot tendency curves of the error rate w.r.t $m$ in Fig.6 on 4 real-world datasets. From this figure, we get similar results as we discussed before.

Figure 6: *Tendency curves of error rate of learning* `AoRR` *loss w.r.t. m on four datasets.*

Figure 7: *The class-wise error rates of two methods with different noise level data.*

## C.3 Performance of Additional Evaluation Metric on `TKML`

We also adopt a widely used multi-label learning metric named average precision (AP) for performance evaluation. It is calculated as [39]

$$AP = \frac{1}{n} \sum_{i=1}^{n} \frac{1}{|Y_i|} \sum_{j \in Y_i} \frac{|\{\tau \in Y_i | rank_f(x_i, \tau) < rank_f(x_i, j)\}|}{rank_f(x_i, j)}$$

where $rank_f(x_i, j)$ returns the rank of $f_j(x_i)$ in descending according to $\{f_a(x_i)\}_{a=1}^{l}$.

From Table 11, we can find our `TKML` method outperforms the other two baseline approaches on all datasets. For the Emotions dataset, the AP score of `TKML` is 2.16% higher than the LSEP method and near 10% higher than the LR. The performance is also slightly improved on Scene and Yeast datasets. These results demonstrate the effectiveness of our `TKML` method.

| Datasets / Methods | Emotions | Scene | Yeast |
|---|---|---|---|
| LR | 74.85 | 71.6 | 73.56 |
| LSEP | 82.66 | 85.43 | 74.26 |
| TKML | **84.82** | **86.38** | **74.32** |

Table 11: *AP (%) results on three datasets. The best performance is shown in bold.*

## C.4 Performance on Each Class of the MNIST for Effects of `TKML`

**Performance on each class.** To evaluate our method is better than $SVM_\alpha$ on the noisy data, we plot the class-wise error rate w.r.t different noise level data. As seen in Figure 7, our method `TKML` outperforms $SVM_\alpha$ on the flipping classes such as 2 and 3, especially in class 5. As the noise level increases, the performance gap becomes more pronounced. For flipping class 7, the performance in this class is increased when the noise level increases from 0.3 to 0.4. The flipping class 6 also get good performance on the noise level 0.2 and 0.3.