[Reviews · NeurIPS 2020]

Review 1

Summary and Contributions: Update after reading the author rebuttal I'd like to thank the authors for their detailed rebuttal, and I appreciate that they have included additional experiments. I will briefly reply to some of the points in the rebuttal: - I agree that the proposed AoRR loss is a general approach to handling outliers, and that existing approaches such as Huber loss and capped hinge loss are individual losses. There is no question that the AoRR loss allows for general applicability because it doesn't require changing the individual loss. However, this should be discussed in depth in the manuscript: if the problem that SoRR addresses is one of learning with robustness against outliers, existing approaches should be presented and their pros and cons need to be considered (and perhaps can also be compared in the experiments). - In my opinion the comparison with the maximum loss is still presented unfairly in the manuscript, in the sense that it is presented as if [27] do not recognize the problem of outliers. I doubt anyone uses the maximum loss in the way in which it is used here (AoRR with m=0, k=1), and including it as such in the tables thus leads to an unfair comparison (in the sense that it presents an improvement over a method that isn't used in practice, and that w.r.t. outliers is known not to work well). - On the comparison with ATk I agree that there will be no benefit to AoRR if there are no outliers, but using AoRR in practice would still incur significant additional computational costs. Also, while the differences on Monk and Phoneme do indeed appear to be significant, it is worth pointing out that the standard deviation of a difference is computed as sqrt(sigma_1^2 + sigma_2^2), not as the authors write in their rebuttal, as sigma_1 - sigma_2. - The rebuttal does not alleviate my concerns regarding the computational costs of the proposed method. Not only is optimization more expensive than average loss for a single setting of k and m, these parameters potentially need to be optimized to get the best results. To give the reader a proper understanding of the advantages and disadvantages of the proposed method, this should be contrasted with the computational cost of existing robust learning algorithms. --- This work introduces a novel aggregate loss function for binary and multiclass/multilabel classification. Instead of taking the average or maximum of individual object losses, the authors propose to use the average of a limited set of object losses that are among the largest but exclude the largest ones, to be more robust against outliers. This manuscript proposes to optimize the resulting aggregate loss function using the difference of convex algorithm, and presents experimental results for both binary and multiclass/multilabel classification.

Strengths: The strengths of this manuscript lie in the detailed execution of the work as well as the favorable experimental results. It is clear that outliers can have an effect on a learned decision boundary and by adjusting the aggregate loss the authors give a straightforward way to deal with this problem in a model-agnostic fashion. While there has been some work on alternative aggregate loss functions, this work gives a relatively novel approach and demonstrates that it performs well in practice. Therefore I believe this work could have an impact on the broader community.

Weaknesses: My main concerns for this work are with the clarity of the description and the motivation for the method. In the introduction the authors discuss other aggregate losses such as the average, maximum, and average top-k losses, but do not discuss the many other ways that have been devised to deal with outlying observations (such as instance weights, Huber loss, etc.). A clear motivation of why the proposed aggregate loss might be preferable over commonly-used alternatives is necessary to be able to place this work into context.

Correctness: The theoretical claims of the paper are accompanied by detailed proofs in the supplementary material. I have briefly checked the proofs and at first glance they appear to be correct. With regards to the experimental evaluation of the work, some improvements can be made. First, the use of the maximum loss is done somewhat unfairly. On line 111 it is defined as the maximum individual loss and the work of Shalev-Swartz & Wexler (2016) is cited as reference. However, in that work the fact that the maximum loss is sensitive to outliers is recognized and this is addressed with a robust variant of the maximum loss. It seems that the authors of the present paper have chosen not to use this robust variant but instead use the naive maximum loss (which is trivially not robust against outliers). Second, while the experimental results on the real datasets show good performance of the proposed method, in many cases the performance of the average or AT_k loss lies within less than one standard error of the best performance. This suggests that the experiments are insufficient to support the claim of better performance for AoRR with any reasonable statistical significance. Finally, the comparison in Figure 3 seems unfair towards the AT_k aggregate loss: setting k = 2 ensures that the AT_k loss likely receives the loss for the outlier and for one other point. The examples in Figure 1 of the AT_k paper (which, incidentally, show a striking resemblance to those in Figure 3), clearly use a larger value of k (k = 10) for these examples.

Clarity: In terms of language and structure the paper is clearly written and easy to follow. However, some important details are missing from the description. For instance, the complexity of the DCA algorithm is not discussed in the main text. This is given in the supplementary material and seems quite significant when compared to the average loss (especially for larger datasets). In addition, the experimental results show that the parameters k and m of the AoRR loss need to be optimized using a grid search (with many possible choices for m). It thus appears that there is a nontrivial computational burden to using the proposed loss function. This is however not discussed in the text. Furthermore, lines 209-215 discuss a method for choosing k that can be used in practice, but the subsequent experiments appear to optimize k and m using a grid search. This raises the question of how feasible the proposed method is in practice.

Relation to Prior Work: As discussed above, the motivation for this work could benefit from a broader overview of related work on addressing outliers in classification, and why the authors believe that changing the aggregate loss is the preferred method. Moreover, the use of the difference-of-convex algorithm in combination with a top-k loss to address outliers has been discussed previously in [1], albeit in the context of SVMs. [1] Chang, Yu, & Yang, "Robust Top-k Multiclass SVM for Visual Category Recognition", KDD 2017.

Reproducibility: Yes

Additional Feedback: Minor comment: * The authors of the AT_k paper show in their Figure 1 how the misclassification error evolves with the choice of k. Such figures could be informative for the AoRR method as well, perhaps showing the evolution with different values of k and m. This could potentially be incorporated in the supplementary material. Note on the broader impact statement: * The authors state that a broader impact statement is not applicable for their work. While I mostly agree with this considering the theoretical nature of the work, I can also imagine practical scenarios where outliers are informative of problems in the data. In such cases a loss function that ignores these outliers may hide information, which may have negative downstream consequences.


Review 2

Summary and Contributions: The paper proposes a new class of aggregate loss functions applicable to a variety of classification problem settings. Popular instances in this family, e.g. average top-k loss, have been studied recently, and the paper proposes a simple and intuitive generalization. Experimental findings on different problem settings and datasets are supportive. I like the paper overall, but I'm not quite sure of the novelty of the technical contributions in this work.

Strengths: The proposed generalization SORR is simple, intuitive and admits a difference of convex program formulation. This enables DC programming solutions to the problem, which have convergence to global optimum guarantees under certain assumptions (this part is not addressed in the paper though). I like that the paper shows loss instances and results on two different problem settings and the demonstrated results are supportive of the claims in the paper; it also encourages using other instances from this family that is not currently explored as alternatives to consider by machine learning practitioners and well as learning theoreticians.

Weaknesses: I found the novelty of the work to be somewhat limited - which is still okay, I find the range of problems and loss functions shown to be fairly sufficient and interesting for the NeurIPS and ML audience; but the paper doesn't make any significantly new technical contributions (for e.g. Theorem 1 appears like a corollary to the result in [10], which already makes this simple and crucial connection). -----Update----- I've read the response. My opinion/score remains unchanged. -----------------

Correctness: I believe the results and methodology are correct. I've not fully verified the proofs.

Clarity: Yes, the paper is quite clearly written.

Relation to Prior Work: Yes.

Reproducibility: Yes

Additional Feedback: I don't have anything in particular to add, beyond my comments above, with respect to what I like about the work and where I find it a bit lacking. There's one comment on the experiments: did the authors consider a simple baseline that removes outliers in an unsupervised fashion based on just the features, and then try a method like average top-k? It would be good to see what the proposed loss function achieves over and top of this simple baseline; this will also strengthen the case for using the proposed loss in practice.


Review 3

Summary and Contributions: This paper introduces the sum of ranked range (SoRR) as a general learning objectives, the minimization of which can be solved with the difference of convex algorithm (DCA). Moreover, this paper also shows how to solve binary classification and multi-label/multi-class classification by the minimization of the SoRR objectives.

Strengths: This paper claims that the proposed SoRR can alleviate the influence of outliers which is validated by comparative studies. Both the motivation and the proposed method are OK. The idea generalizes some traditional loss functions (e.g., average loss, maximum loss, median loss, average top-k loss, etc.) and might be effective with proper parameter settings.

Weaknesses: 1. In my opinion, the major motivation of this paper is stated in the 2nd paragraph of Introduction, i.e., "The average is insensitive to minority sub-groups while the maximum is sensitive to outliers". In Subsection 3.2, authors generate two sets of synthetic data (Fig.3) to show the robustness of the proposed method against outliers. Throughout this paper, baselines only include three variants of SoRR, i.e., Average, ATk, Maximum, while there are many loss functions which are designed to alleviate the influence of outliers, such as the capped hinge loss in the following reference: F.-P. Nie et al. "Multiclass capped ℓp-Norm SVM for robust classifications." In AAAI, 2017. Moreover, such outliers shown in Fig.3 actually can be regarded as label noise, i.e., the inaccurate supervision case in the following reference: Z.-H. Zhou. "A brief introduction to weakly supervised learning." In NSR, 2018. 2. My another concern is whether fair comparison is conducted in experiments. Besides the baseline selection, the parameter setting is also very critical. Authors might intentionally give a set of parameters which is in favor of the proposed method in experiments. For example, authors select k=2 in Fig.3, which might be a better parameter for AoRR, but not an appropriate one for ATk. Similiar problems also exist in other experiments. **I've read the authors' responses, which clarified my concern regarding the choice of k in the experimental studies.**

Correctness: The idea generalizes some traditional loss functions (e.g., average loss, maximum loss, median loss, average top-k loss, etc.). With proper parameter setting, the proposed method outperforms the selected baselines, but I have some concerns here which have been stated in Weaknesses above.

Clarity: This paper is well written and easy to follow. Besides, for the definitions in the 1st paragraph of Section 2, it would be better to give an illustrative example similar to Fig.1.

Relation to Prior Work: The proposed method can alleviate the influence of outliers, which is validated by the reported experimental results. However, some related works, such as capped hinge loss, learning with inaccurate supervision, are not discussed in this paper.

Reproducibility: Yes

Additional Feedback: See details in Weaknesses.

[Author Response · NeurIPS 2020]

We thank all reviewers for the constructive comments. Due to the space limitation, we only address major review
concerns and will incorporate other suggestions in a revised version of the current work.

**R1-Q1: Motivation of using the SoRR loss**. Traditional approaches to handle outliers focus on the design of robust
*individual losses* that apply to individual training samples, notable examples include the Huber loss and the capped
hinge loss as pointed out by the reviewer, or give pre-determined weights to individual training sample. We take a
different approach to this work. Instead of changing the definition of individual losses, our method builds in robustness
at the aggregate loss level, using the AoRR loss. The resulting learning algorithm is more flexible and allows the user to
choose an individual loss form that is relevant to the learning problem. On the other hand, the weights on the individual
training sample under AoRR loss is determined automatically.

**R1-Q2: Related work of (Chang, Yu, & Yang, KDD 2017)**. This work addresses the outliers for multiclass SVM by
using a hyperparameter to cap the values of the individual losses. This approach is *different* from ours since we directly
address the original top-$k$ multiclass SVM problem using our TKML loss *without* introducing this convex surrogate.
Furthermore, it has been shown that the loss used in this work is not multi-class top-$k$ consistent as shown in [34], while
our TKML loss is consistent (see lines 265-270).

**R1-Q3: Comparison with maximum loss**. Although the sensitivity to outliers of the maximum loss can be alleviated
with the tricks used in [27], our work on the AoRR aggregate loss provides a general mechanism to exclude the influence
of potential outliers in the training data. We use the straightforward maximum loss, as it is a special case of the AoRR
loss ($m = 0, k = 1$) and can better reflect the continuum of performance change while we adjust $k$ and $m$. As such, we
do not want to give any special treatment to this special case of AoRR loss.

**R1-Q4: Performance comparison with the $AT_k$ loss**. The advantage of the AoRR aggregate loss is its improved
robustness to outliers. On datasets that do not have a significant fraction of outliers, its performance is expected to be
similar to the $AT_k$ loss. On the other hand, when the dataset contains outliers, AoRR can significantly outperform
existing aggregate losses, as demonstrated on datasets Monk and Phoneme (Table 1, with an average difference/standard
deviation in the accuracy of 4.07%/0.05% and 2.22%/0.18%, respectively).

**R1-Q5: Value of $k$ when comparing with the $AT_k$ loss**. Experiments on the synthetic
data aim to show that the AoRR loss can exclude the influence of outliers. To exemplify
this case, we chose to have one outlier in the synthetic data, and correspondingly, use
$k = 2$. Furthermore, we perform an additional experiment to the request of R1 showing
misclassification rate w.r.t different values of $k$ in Figure 1. This result shows that for
values other than 2, the AoRR loss still exhibits an advantage over the $AT_k$ loss.

**R1-Q6: Computation complexity of optimizing AoRR**. For large datasets, the orig-
inal DC algorithm may take a longer time to run. However, we can use a stochastic
version of DCA to optimize our problem according to Thi, Hoai An Le, et al. "Stochastic
DCA for minimizing a large sum of DC functions with application to Multi-class Lo-
gistic Regression." arXiv preprint arXiv:1911.03992 (2019). The authors have proved
it is much more efficient and the convergence of it to a critical point is guaranteed with
probability 1.

Figure 1: *Misclassification rates
of the $AT_k$ loss (blue) and AoRR
loss (red) for different $k$ values
on a balanced but multi-modal
synthetic dataset.*

**R1-Q7: Grid search for $m$ and $k$**. Our experiments are based on a grid search for
selecting the value of $k$ and $m$ which may not be ideal in practice for large-scale
datasets. For this reason, we provided a well-know adaptive setting method there, and its feasibility had been discussed
in [13]. Since $k$ and $m$ are hyper-parameters which are chosen by cross-validation (CV), we did not include the time of
searching for them into the overall time complexity, as commonly practiced in machine learning.

**R1-Q8: Strategy to choose k**. We suggested a strategy to choose $k$, but for the experiments we performed, we found
that a grid search is simpler and often yields comparable performance.

**R2-Q1: Novelty of this work.** Although Theorem 1 is related with Lemma 1 in [10], there are some fundamental
differences. The average top-$k$ function in [10] is convex with regards to its inputs. In our work, we show that the SoRR
can be expressed as the difference of two sum of top-$k$ functions, which shows that it is not convex but can be solved
with the DC algorithm. This connection has not been studied previously. In addition, based on the SoRR approach, we
describe in details of new types of aggregate loss that are more robust to the presence of outliers in training data, and
new types of individual losses for multi-class multi-label learning.

**R4-Q1: Related work of (Nie, et.al., AAAI 2017).** The capped hinge loss in this work is designed as an individual
loss. Since the AoRR loss is at the aggregate level, we did not compare it as related work.

**R4-Q2: Related work of (Z. Zhou, NSR 2018).** The outliers in Figure 3 are different from the case of inaccurate
supervision as described in (Z.-H Zhou, NSR, 2018), which assumes errors in the labels of training data. In our study of
aggregate losses, we assume the training labels are *accurate* for all training samples, including the outliers.

**R4-Q3: Value of $k$ when comparing with the $AT_k$ loss.** Please refer to our answers to R1-Q5.

[Meta-Review · NeurIPS 2020]

The paper introduces a novel aggregate loss function for binary and multiclass/multilabel classification to obtain models being robust against outliers. The reviewers are, in general, positive about the paper, however, there are some flaws making the paper a borderline case. The paper should discuss in depth the relation to the classical approaches and perhaps use them in the experimental studies. The comparison with the maximum loss seems to be slightly unfair as the authors use its vanilla variant. The complexity of the approach is not clearly discussed.